# The Effects of Core Stabilization Trunk Muscle Fatigue on Lower Limb Stiffness of Basketball Players

**DOI:** 10.3390/sports11100200

**Published:** 2023-10-12

**Authors:** Mehdi Khaleghi Tazji, Hassan Sadeghi, Ali Abbasi, Mohammad Aziminia, Ali Shahhosseini, Mohammad Ebrahim Marjani, George A. Koumantakis

**Affiliations:** 1Department of Biomechanics and Sports Injuries, Faculty of Physical Education and Sport Sciences, Kharazmi University, Tehran 1544733111, Iran; mehdikhaleghi60@yahoo.com (M.K.T.); abbasi.bio@gmail.com (A.A.); m.aziminia1510@gmail.com (M.A.); 2Department of Coaching, Faculty of Physical Education and Sport Sciences, Kharazmi University, Tehran 1544733111, Iran; shahhosseini_ali@yahoo.com; 3Department of Physical Education, Shahid Chamran Campus, Farhangian University, Tehran 1939614464, Iran; mohammadebrahimmarjani@gmail.com; 4Laboratory of Advanced Physiotherapy, Physiotherapy Department, School of Health and Care Sciences, University of West Attica, 12243 Athens, Greece; gkoumantakis@uniwa.gr

**Keywords:** core stability, athletic injury, lower extremity stiffness, vertical jump, basketball player

## Abstract

Core stability is the ability to control the position and motion of the trunk over the pelvis and legs to allow the optimum production and transfer of force to the terminal segment in sporting activities. The effect of fatigue, especially core muscle fatigue, on stiffness as a performance index requires more study. This research aimed to investigate the effect of the core stabilization muscles’ fatigue on lower limb stiffness during hopping. Thirty active basketball players participated in this study (age: 20.90 ± 1.49 years; weight: 60.30 ± 3.10 kg; height: 163.2 ± 5.04 cm). The hopping test (15 jumps) was performed before and after the fatigue protocol in three states including at a preferred (no frequency control), maximum, and 2.2 Hz frequency on the force plate. The stiffness of the lower extremities was measured before and after the fatigue protocol. The results of the dependent t-test showed core muscle fatigue led to reduced lower extremity stiffness under all three hopping-test conditions by 15.3–15.9% (*p* ≤ 0.005). It seems that core muscle function affects lower extremity stiffness, and can function as a performance index in athletes. Although performed in healthy volunteers, this study may have implications for injury prevention, highlighting the necessity to perform interspersed endurance training using the different body parts of the kinetic chain.

## 1. Introduction

A loss of core stability can expose one to the risk of injury and proper training can reduce the probability of injury [1]. In addition, the weakness or loss of core muscle coordination can cause abnormal movement patterns, compensatory movement patterns, or various types of sports injuries such as muscle strain and overuse injuries [2]. Since core muscles are essential for carrying out appropriate movements by creating a stable base of support, the fatigue of these muscles may affect the performance of an individual, especially basketball players [3,4]. A study by Surenkok et al. (2008) examined the effect of abdominal muscle fatigue via an isokinetic protocol, and the accumulation of lactic acid, on the static and dynamic balance of healthy individuals, concluding that muscle fatigue had a negative effect on the static and dynamic balance of the subjects [5]. The effect of the fatigue of the more centrally located gluteus medius on the neuromuscular performance of the lower extremity has been evaluated [6]. The results showed that gluteus medius fatigue caused a significant reduction in men and women’s static and dynamic balance and their quality of movement, but there was no difference in the rate of reduction between men and women. In addition, a study on basketball players demonstrated that muscle fatigue has negative effects on basketball lower-leg-muscle activity changes and leads to an increased risk of injury and reduction [3].

In the healthcare sector, performing daily activities without pain is desirable, but in the field of competition, achieving the best performance with the lowest risk of injury is a concern [7]. Due to the neuromechanical properties of human movements [8], achieving this goal requires an understanding of neuromuscular aspects and the mechanical properties of body structures [9]. One of the mechanical properties of viscoelastic structures, which is the scale of their resistance to deformation, is called stiffness [10]. The function of the body in some motions can be considered to be like a spring that, with its compression, saves energy and in the following period, gives it back again.

Despite the complex dynamics of the neuromuscular system, simple mass-spring models can acceptably describe these movements [11]. Stiffness is conducive to adjusting the Rate of Force Development (RFD), power-saving capability of the Stretch-Shortening Cycle (SSC), and the amount of system resistance against incoming disturbance [12]. It has been reported that high stiffness increases the amount of force development, which becomes useful for speed performance [13]. In a performance like a vertical jump, during which a height is recorded and time does not determine the criterion for the record, excessive stiffness may prevent the proper storage and reproduction of energy [14]. Tendon and muscle stiffness are measured in the laboratory using direct methods. The weakness of laboratory methods is that research conducted on a specific muscle or tendon does not provide a panoramic view of the movement. So, in the laboratory, for instance, we inevitably need to monitor the change in the length of the Achilles tendon when calf muscles isometrically contract, but this shows only a specific part of, e.g., basketball players [15].

Several methods under natural conditions have been devised for determining stiffness, which measure this variable at the lowest level (stiffness of muscles and tendons), middle level (stiffness of a joint and its surrounding structures), and highest level (stiffness of the limb or the entire body). Various methods have been presented to determine stiffness in biomechanics research. One of these methods is the mass-spring model, which is capable of describing some of human motion. The stiffness obtained from this method is a combination of stiffness values from all body structures and can be influenced by muscle activity levels, nerve reflexes, and individual strategy in the movement [16]. The most simple mass-spring model consists of a linear spring without mass and a point mass that represents the center of gravity of the whole body. The stiffness of this spring while hopping and running is a significant parameter for determining its center-of-mass mechanics [17].

A wide range of research in this field has given attention to studying mechanical stiffness (stiffness of the leg and lower extremity) during exercises such as running, walking, and hopping. Hopping is a biomechanical movement that mimics the spring-like motion of kangaroos and is commonly employed as a clinical test to evaluate the mechanical stiffness and elastic properties of muscles and tendons in the lower extremities [14,18]. It seems that core muscles play an important role in the mechanics of the lower extremity for the injury prevention and performance improvement of basketball players. To the best of our knowledge, far too little attention has been paid to evaluating the effects of core muscle fatigue on the lower extremities of Basketball players. The hypothesis that will be tested in this paper is that core stability muscle fatigue can affect the stiffness of lower-extremity structures. Therefore, the aim of this study is to investigate the effect of core stabilizer muscle fatigue on the stiffness of the lower extremities of basketball players.

## 2. Materials and Methods

### 2.1. Participants and Study Design

Thirty young male basketball players (age: 20.90 ± 1.49 year, body mass: 60.30 ± 3.10 kg, body height: 163.2 ± 5.04 cm, BMI: 21.79 ± 1.71 kg/m^2^) participated in this quasi-experimental study design. Participants were invited to the laboratory and the implementation process of the tests was explained. Participants were recruited via announcements to university basketball players.

This study protocol was approved by the Department of Biomechanics and Sports Injuries research committee, Faculty of Physical Education and Sports Sciences, Kharazmi University, Tehran, Iran (Approval ID: KHU/PESS.1401.2547 in December 2022). The participants were informed about the details of the study and provided their written informed consent before study enrolment. Informed consent was obtained from all the participants and procedures were conducted according to the Declaration of Helsinki. Participants were assessed for laterality and were right-side dominant for all criteria, which was identified by using the ball kicking test. The inclusion criteria consisted of age range of 18 to 25 years, no history of injury in the past six months, and experience in playing basketball for at least 3 years. The exclusion criteria consisted of neuromuscular disorders or cardio-vascular problems (Figure 1).

### 2.2. Procedure

At first, anthropometric characteristics including the weight, height, and length of the subjects were measured and recorded. The participants were taught how to hop. The hopping test was performed in three states including at a preferred (without frequency control, height, or contact time), maximum, and 2.2 Hz frequency. Subjects performed a warm-up protocol (including 5 min running at their desired speed and 5 min static stretching), before the hopping test. In the preferred hopping test, the subject performed arbitrary hopping. The jump height and frequency were not controlled. In the 2.2 Hz hopping test, a metronome was used and the subjects were asked to coordinate their jumps with the metronome rhythm. Finally, subjects performed a hopping test with a maximum jump height and minimum contact time. All three tests included these considerations: The hopping test was performed vertically and each test was constructed of 15 consecutive jumps. To reduce the effect of the upper limb and shoes on the hopping test, subjects’ hands were put on their waists and they were barefoot. All three hopping tests were performed on both legs, with each leg acting as a pair, to evaluate the mechanical stiffness and elastic properties of muscles and tendons in the lower extremities. The subjects were allowed to practice to ensure the proper implementation of the hopping.

### 2.3. Fatigue Protocol

The fatigue protocol developed by Abt et al. (2007) was used to measure core stability muscle fatigue in multiple planes of movement [19]. Benjaminse et al. (2008) had proved the validity and reliability of this protocol in central muscles through isokinetic strength testing [20]. The protocol lasted approximately 30 min and consisted of four sets of exercises. Each exercise was performed 20 times for 40 s, with a 20 s rest period between each workout. A complete set comprised the following movements: seated upper-torso rotations with a medicine ball, static prone torso extension with a medicine ball, supine lower-torso rotations with a medicine ball, incline sit-ups with a weighted plate, lateral side bend (performed bilaterally) with a weighted plate, rotating lumbar extension with a weighted plate, and standing torso rotations with weighted-pulley resistance. To ensure core muscle fatigue, the subjects were asked to continue exercise to the point of failure. According to previous studies, two criteria were used to define the core muscle fatigue [19,20]: (1) when the subject was unable to perform the exercise in the fourth set properly or with correct form, and (2) when the subject was unable to perform the exercise with a repetition rate of two seconds in the fourth set. When one of the two above-stated criteria occurred in during the performance of each of seven core workouts in the fourth set (the final set), the protocol was discontinued and the subjects were given 20 s of rest. The subjects performed the next workout to fatigue again until one of the criteria would happen, then rested for 20 s. This process was continued in the seventh exercise in the fourth set. Immediately (less than 10 s) after the occurrence of fatigue in the seventh exercise, a lower-limb stiffness test was conducted [21].

### 2.4. Test Procedure

Force plate (AMTI model, 40 × 60 cm, Bertec, UK) ground reaction force (GRF) data during hopping in the pre- and post-test was recorded at 1000 Hz. The data were filtered with a zero-lag fourth-order low-pass Butterworth filter at 50 Hz [9]. The GRF data from the sixth to the tenth jump (of fifteen jumps) were selected. Leg stiffness (*K_leg_*) was calculated using the equation of McMahon et al. (1990) [22]. The *K_leg_* was calculated by dividing the maximum vertical ground reaction force (*F_max_*) by the downward displacement of the center of mass (Δ*y*) during the ground-contact phase (Equation (1)). The *F_max_* contact data were extracted directly from force plate data. Various methods have been proposed to calculate Δ*y*. In this study, using the subject’s mass and GRF, the acceleration was calculated. Finally, Δ*y* was determined through double integration of the acceleration data.
(1)$$K_{leg}=\frac{F_{max}}{\Delta y}$$

The mechanical stiffness for each of the five jumps (jumps six to ten) was calculated and the average of these values was normalized to the subject weight. Figure 2 shows vertical GRFs in jumps six to ten for a subject. To compare the amount of stiffness before and after the fatigue, a paired *t* test with a significance level of α ≤ 0.05 was used.

### 2.5. Statistical Analysis

Statistical analysis was performed using SPSS software (version 22). Descriptive statistics were used to analyze baseline data for demographic data. The Shapiro–Wilk test results indicated that all data were normally distributed. A paired-samples *t*-test was used to find the significant differences between the pre-test and post-test. Due to the sequential use of the paired-samples *t*-test, the Bonferroni correction was used to counterbalance the multiple testing applied [23]; therefore, the α level was set to 0.05/3 = 0.017.

## 3. Results

The results of the dependent *t*-test showed that core muscle fatigue significantly reduced lower-limb stiffness during preferred hopping by 15.33%, 15.9%, and 15.44% compared to the preferred (t29 = 5.47, *p* = 0.000), 2.2 Hz (t29 = 4.78, *p* = 0.005) and maximum hopping (t29 = 4.17, *p* = 0.000), respectively. Mean and standard deviation of stiffness during preferred, 2.2 Hz, and maximum hopping in pre- and post-test core muscle fatigue reported in Table 1. The stiffness during the preferred, 2.2 Hz, and maximum hopping in the pre- and post-core muscle fatigue are shown in Figure 3.

## 4. Discussion

We assumed that core muscle fatigue leads to changes in the stiffness of the lower extremity during the performance of hopping. The results of the study show a reduction in the stiffness of the lower extremity during all three types of hopping following core muscle fatigue. Because many lower-extremity injuries, especially those of the ankle and knee, occur during landing, the hopping maneuver was selected. In performing functional skills that are accompanied by weight loading, stiffness can be affected by muscles, tendons, ligaments, and the skeletal alignment, which works as a unit to modify the rotational stiffness of a joint while making contact with the ground [18]. The rotational stiffness is controlled by neuromuscular and biomechanical factors consisting of the activation of the muscle and force, reflexes, the simultaneous contraction of antagonist muscles, and kinematics of the lower extremity while in contact with the ground [20].

Given that fatigue influences these factors, it seems that stiffness also changes as a result of muscle fatigue. The changing of control strategies caused by fatigue can affect the probability of the risk of lower-extremity injury. It is still unknown and requires further investigation how core muscle fatigue is conducive to changes in lower-extremity stiffness. Given that the core muscles are essential to create a stable surface for the proper movement of the upper and lower extremities, core muscle fatigue can affect sports activities [24]. Biomechanical and neuromuscular factors such as muscle activation patterns, co-contraction, and kinematic, kinetic and stiffness characteristics change as a result of fatigue [24].

It has been reported that the stiffness of the lower extremity has a significant role in improving the performance of running, jumping and hopping exercises, which are seen in many sports. Athletes who use stiffness characteristics more efficiently save more elastic energy during landing, which consequently results in a more concentric force in the push off. This state delays fatigue and ultimately increases running speed [25]. The relation between stiffness and performance is polyhedral. Many studies have reported that stiffness of a lower extremity increases with hopping height and running speed. It has been reported that foot conditions (increase in the Range of Motion, the length of muscles and tendons, and soft tissues) can improve running and jump performance, while leg stiffness remains constant or even reduces [26,27]. On the other hand, the relation between the stiffness and the risk of injury has been reported as being positive: this means that too much increase in the stiffness can increase the risk of injury. Nevertheless, a direct correlation between the stiffness of the lower extremity and lower-extremity injuries has not been made clear, perhaps because of the few studies that have been conducted in this area. Too much stiffness of the lower extremity is related to a reduction in joint mobility and an increase in the shock and force of lower extremity; on the other hand, too little stiffness is accompanied by high joint mobility.

Core stabilization depends on the ability to control the body in response to internal forces and external disturbances. These forces consist of forces produced from the distal parts of the body as well as expected and unexpected external disturbances [28]. It is believed that the trunk muscles play an important role in balance and bring about improvements in body control and balance with an increase in core strength among basketball players [29]. It has been shown that the activation of upper- and lower-extremity muscles under different speed conditions affects the core muscles [28,30]. For instance, Hodges et al. (1997) examined muscle timing during body movements and found out that a number of core stabilizer muscles (such as transverse abdominis, multifidus, rectus abdominis, and oblique abdominis) contract before lower-extremity movements [31]. Asymmetry in proximal muscle activation and a reduction in the activation of trunk and hip muscles might reduce the potential of appropriate muscular activation patterns in response to the joint load [32]. It has been shown that proximal muscle fatigue in comparison to distal muscle fatigue is conducive to more reductions in postural control. One possible reason is that the proximal muscles (such as hip muscles) in comparison to the distal muscles (such as the muscles of the ankle) have a larger cross-sectional area and, thus, these muscles can be more effective against fatigue [31]. When a person makes an effort to maintain his or her posture status, modified contractions permanently occur in response to small disturbances of joints.

Alternatively, fatigue can be considered as a generalized and not only a localized phenomenon. A recent systematic review lends some experimental support to the generalized non-local muscle fatigue effect, particularly when examining specific types of performance outcomes, like the hopping tasks examined in this study [33].

Because fatigue reduces the speed of neural transfer, the ability to produce compensatory contractions around joints might be reduced and is conducive to weaknesses of neuromuscular control and further changes in the joint’s condition. Also, more variability in the range of joint movement might reduce the balance in the absence of corrective muscle actions [34]. Given that the stiffness test is performed in the closed kinetic chain, changes produced by fatigue in proximal parts affect the distal parts and cause a reduction in stiffness. It seems that the performance of the core muscles has an effect on the amount of stiffness. Decreases in the values of stiffness following core muscle fatigue during preferred hopping, hopping with 2.2 Hz frequency and maximum hopping were 15.33%, 15.09%, 15.44%, respectively. The used 2.2 Hz frequency is known as the optimal frequency, because at frequencies below 1.5 Hz, the springing property does not activate, and frequencies above 3 Hz also cause a loss of elasticity [26]. Most researchers have confirmed 2.2 Hz frequency to be ideal in order to examine hopping [34,35,36]. Nonetheless, some researchers believe preferred hopping (without controlling the frequency or height of jumps), and others believe maximum hopping (with minimum time contact and maximum height) to be the appropriate hoping test. Due to this disagreement, we decided to examine all three types of hopping and compare the extracted results. According to the test results, we can generally conclude that the function of core muscles affects the amount of stiffness.

The results of the present study have shown the effect of the fatigue of core stabilizing muscles on the stiffness of the lower extremity during hopping, which can be used to enhance training foundations for athletic performance, injury prevention, and rehabilitation. This significant topic has highlighted injury prevention methods and aided towards enhancing the quality of performances of athletic protocols, and has highlighted the necessity to pay more attention to the better quality of training components. According to the advent of more effective spring-like hopping over running, perhaps it is possible to use stiffness in hopping as a more proper index for the level of elastic properties of the muscular system. During hopping, the foot stiffness is equal to the vertical stiffness, and it is extracted by dividing the vertical force of the ground by the displacement of the center of mass. Active muscular stiffness is the main factor in the modification of disturbances and the establishment of dynamic stability. The deterioration of this factor causes the growth of disturbances and places the joint at the end of the ROM, which causes an increase in the tension of ligaments and a risk of injury. The required stiffness for achieving the highest level of performance in sports activity might put the athlete at risk of injury. So, we have to consider an optimum level of stiffness that caters for injury-free sports participation, while maintaining a high level of performance at the same time.

## 5. Limitations

This study provided some valuable insights into the effects of core stabilization muscle fatigue on the lower-limb stiffness of basketball players, but it had several limitations. First, it only included male basketball players. Since female athletes have a higher injury risk in similar activities than men, future studies should investigate the gender effect. Second, in addition to fatigue, other factors that can affect the stiffness of the lower limbs of athletes, such as diet and training, should be considered. Finally, in order to better identify the traumatic factors caused by lower-limb stiffness in athletes, future studies should record the activity of working muscles through electromyography as well as kinematic variables of the lower limb.

## 6. Conclusions

According to the results of this study, there was a significant decrease in the amount of lower-extremity stiffness measured at various frequencies during a fatigue protocol targeting the trunk stabilizing muscles, applied in a group of healthy basketball players. Training programs with an aim to increase the endurance enhancement of the stabilizing muscles of the trunk may lead to less severe deteriorations of lower-limb stiffness, which may improve performance as well as act as an injury prevention mechanism for the safe participation in various athletic activities involving the lower limb.

## Figures and Tables

**Figure 1 sports-11-00200-f001:**
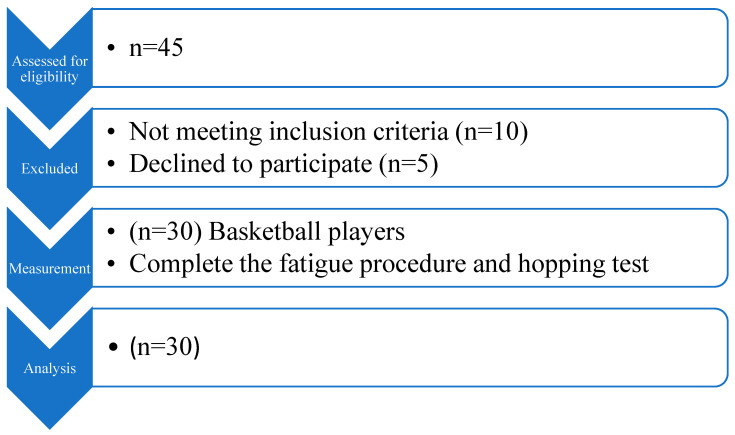
Diagram of the study procedures.

**Figure 2 sports-11-00200-f002:**
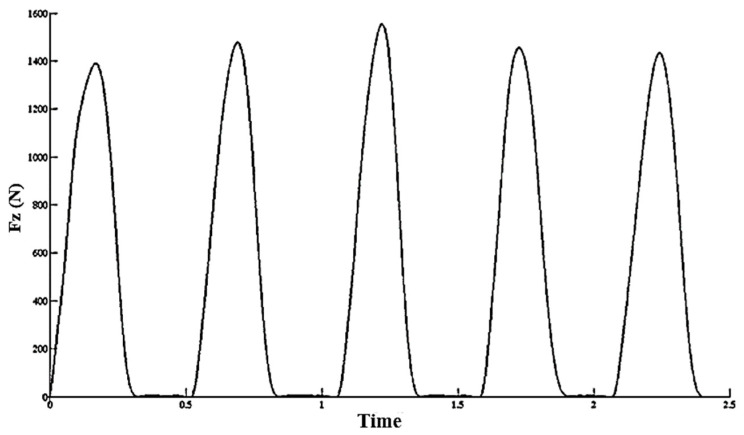
Mean of vertical GRFs during hopping test.

**Figure 3 sports-11-00200-f003:**
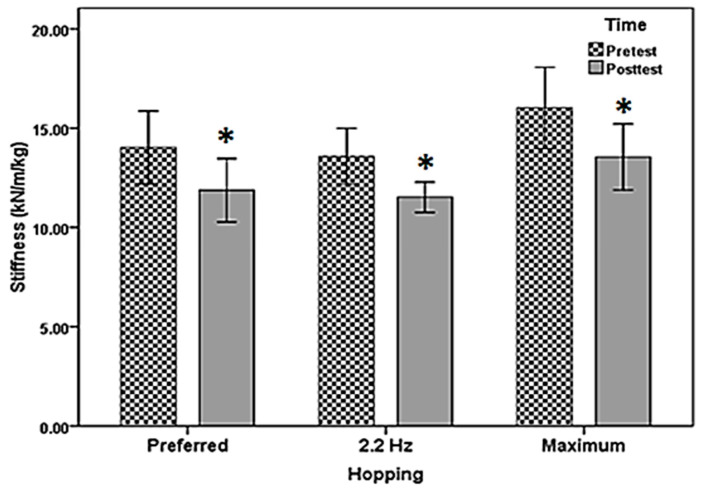
The stiffness during preferred, 2.2 Hz, and maximum hopping in pre- and post (mean and SD)-core muscle fatigue. * Significant differences between pre-test and post-test.

**Table 1 sports-11-00200-t001:** Mean and standard deviation of stiffness during preferred, 2.2 Hz, and maximum hopping in pre- and post-test core muscle fatigue.

Stiffness	Condition (Mean and SD)		*p* Value
	Pre-Test	Post-Test	
Preferred	13.98 ± 2.55	11.38 ± 1.89	0.000 *
2.2 Hz	13.55 ± 1.99	11.60 ± 1.04	0.005 *
Maximum	16.13 ± 2.74	14.22 ± 1.88	0.000 *

* Statistically significant difference (*p* ≤ 0.017).

## Data Availability

The authors declare that they have followed the protocols of their work center on the publication of patient data. Research data will be shared upon request to the corresponding author.

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
