# Peer review of "The Effects of Core Stabilization Trunk Muscle Fatigue on Lower Limb Stiffness of Basketball Players"

_sports, 2023, doi:10.3390/sports11100200_

Round 1

Reviewer 1 Report

The effects of core stabilization muscles fatigue on lower limb stiffness of Basketball players

This is an interesting study in basketball players, which examines the effects of a strong fatigue protocol to vertical jumping performance (three conditions: without frequency control, height, or contact time, maximum and 2.2 Hz frequency). Although the results of the study are promising, there are several points inside the methods which Authors have to clarify. Still the manuscript needs significant revision .

Main concerns:

1. Was there a counterbalanced study design performed? Where the three vertical jump conditions performed randomly?

2. Since the study has a pre and post measurement and three different vertical jump conditions then why a paired sample T-Test was used? And if this is the case, then why α remained at 0.05?

Specific Comments:

Abstract:

Abstract focuses more on the presentation of the problem and less on the presentation of the results and the take home message. I suggest a strong reconstruction of abstract by adding statistical indexes of percentage changes and a solid take home message for readers.

Moreover, hopping is too general term. Maybe changing it to vertical jump performance would be much more suitable for basketball players and for readers.

Introduction:

Intro is a single big paragraph with mixed messages and references. First step Authors have to take is to create paragraphs with clear objectives in order to lead readers to the problem.

Study [3] seems pretty important for the current study. Authors should focus on presenting this study in more detail.

Line 82: Explain the abbreviation COG.

Lines 88-90: Authors wrote “It seems that core muscles play an important role in the mechanics of lower extremity for injury prevention and performance improvement of Basketball players.” Please, add a reference following this sentence.

Line 95: Delete “during performance of the hopping exercise”.

Add the hypothesis of the study.

Methods:

A diagram of the study procedures would be helpful for readers.

Line 97: Only participants are presented here, no study design. Delete the study design.

Provide an age range of the participants. Also, how laterality was evaluated? What playing positions were participants? What was their training experience and whether they were familiar with the vertical jumping test?

Line 107: Please change weight to body mass, height to body height and clarify what the length of subject’s means.

Line 108: Authors wrote: “The participants were taught how to run hopping.” What “run hopping” means?

Line 109: Authors wrote “The hopping test was performed in three states…”. Was there a counterbalanced design of the states? Did all players completed all states?

Lines 113-115: Since a metronome was used why Authors present the condition with Hz and not with beats per minute?

Line 114: Change “your” with “their”.

Line 119: Authors wrote: “All three hopping were performed pairs of legs.” Please, clarify the sentence.

Lines 119-120: I am sure that basketball players with more than 3 years of training experience are familiar with vertical jumping tests. However, Authors should clarify whether a familiarization training session was performed prior to the three jumping conditions or prior to fatigue training protocol.

Fatigue protocol

Authors should clarify the sets, repetitions and rest intervals of the training protocol. Moreover, the fatigue protocol contains exercises with external loads (medicine balls and weighted plates). Define the loads used for each player (load/body mass) and explain how all players underwent the same fatigue prior to the vertical jump testing.

Methodology: How many times subjects performed the fatigue protocol? I assume that this was performed three times equal to the three different vertical jumping tests.

Test procedure

Why only the last 5 jumps were used in the analysis?   

Statistical analysis: Following my second main concern (see above), Authors should add effect sizes and intra-class correlation coefficients (ICCs) for all performance measurements in order to ensure the reliability of measurements.

Results:

Table 1: There seemed to missing units of measurement. Also, Authors wrote “* Statistically significant correlation (P ≤ 0.05).”, but no correlation is presented.  Where there any differences between preferred, 2.2Hz or Maximum conditions?

Figure 1: Is figure 1 presents the same results as table 1?

Discussion:

Discussion has the same problem with introduction. Needs paragraphs (see my comment above).

Is the fatigue protocol functional and real-world for basketball players?  

Lines 177-180: Why only stiffness was measured and not the actual vertical cm height of the jumps? This might have provided further kinematical details into the results.

Line 196: Provide the explanation of ROM before using the abbreviation.

Lines 232-234: This should be in methods.

Limitations are missing. Authors should add a paragraph of limitations.

Discussion is to general and not discussing the findings of the study. A few lines are focusing on the actual explanations of the results.

Conclusions paragraph is confusing, and my suggestion is to be written again focusing on the findings of the study.    

Author Response

We would like to thank the editor and reviewers for their valuable comments; they have improved the quality of our study We diligently addressed the comments and look forward to clarifying further as needed. Manuscript changes in green color and our responses are following:

Main concerns:

  1. Was there a counterbalanced study design performed? Where the three vertical jump conditions performed randomly?

Answer to the reviewer: Yes, there was. This was already mentioned in lines 125-6, where it stated: “In the preferred hopping test, the subject performed hopping arbitrary.”

  1. Since the study has a pre and post-measurement and three different vertical jump conditions then why a paired sample T-Test was used? And if this is the case, then why αremained at 0.05?

Answer to the reviewer: A paired t-test is used when we are interested in the difference between two variables for the same subject. Often the two variables are separated by time. You are correct, a Bonferroni correction could be applied to manage the 3 t-tests, and the new a value would be a=0.05/3=0.017 (highlighted in text-lines 176-7 and footnote of table 1).

The aim of this study was to compare the paired conditions of hopping before and after fatigue. For instance, our objective was to compare preferred hopping with preferred hopping before and after fatigue. Our objective was not to compare preferred hopping with hopping 2 maximum before and after fatigue. Therefore, a pairwise comparison using t-test was employed.

Specific Comments:

Abstract:

Abstract focuses more on the presentation of the problem and less on the presentation of the results and the take home message. I suggest a strong reconstruction of abstract by adding statistical indexes of percentage changes and a solid take home message for readers.

Answer to the reviewer: The Abstract was enhanced, according to your suggestions.

 Moreover, hopping is too general term. Maybe changing it to vertical jump performance would be much more suitable for basketball players and for readers.

Answer to the reviewer: Thank you so much for your attention, we prefer to keep the hopping test because the hop test is a series of functional and quantitative tests used to assess an athlete’s power and strength of the affected to unaffected leg. On the other hand, vertical jump performance is a measure of power and explosiveness. It describes a wide cohort of jump tests in which participants aim to jump as high as possible.

Anyway, if the editor prefers to change we will do it.

 Introduction:

Intro is a single big paragraph with mixed messages and references. The first step Authors have to take is to create paragraphs with clear objectives in order to lead readers to the problem.

 Answer to the reviewer: Yes, correct. This has been addressed, while also trying to maintain a meaningful and sequential line of argument.

Study [3] seems pretty important for the current study. Authors should focus on presenting this study in more detail.

Answer to the reviewer: Thank you for your attention, more information was added to the introduction. Page 2 line 52

Line 82: Explain the abbreviation COG.

 Answer to the reviewer: The abbreviation changed to the full name of Center of Gravity, page 2 line 83

Lines 88-90: Authors wrote “It seems that core muscles play an important role in the mechanics of lower extremity for injury prevention and performance improvement of Basketball players.” Please, add a reference following this sentence.

Answer to the reviewer: The following reference has been added

Sasaki S, Tsuda E, Yamamoto Y, Maeda S, Kimura Y, Fujita Y, Ishibashi Y. Core-Muscle Training and Neuromuscular Control of the Lower Limb and Trunk. J Athl Train. 2019 Sep;54(9):959-969. doi: 10.4085/1062-6050-113-17. Epub 2019 Aug 6. PMID: 31386583; PMCID: PMC6795098.

Line 95: Delete “during performance of the hopping exercise”.

Answer to the reviewer: The sentences was edited

Add the hypothesis of the study.

 Answer to the reviewer: The hypothesis was added to the end of the introduction 

Methods:

A diagram of the study procedures would be helpful for readers.

Answer to the reviewer: The diagram of the study procedures added to the manuscript

Line 97: Only participants are presented here, no study design. Delete the study design.

Answer to the reviewer: The study design added to the sentence

“Participated in this quasi-experimental study design”

Provide an age range of the participants. Also, how laterality was evaluated? What playing positions were participants? What was their training experience and whether they were familiar with the vertical jumping test?

Answer to the reviewer: Corrected. 

Line 107: Please change weight to body mass, height to body height and clarify what the length of subject’s means.

Answer to the reviewer: Corrected. 

Line 108: Authors wrote: “The participants were taught how to run hopping.” What “run hopping” means?

Answer to the reviewer: Corrected. 

Line 109: Authors wrote “The hopping test was performed in three states…”. Was there a counterbalanced design of the states? Did all players completed all states?

 Answer to the reviewer: All participants performed all three types of hopping. The conditions were executed in a counterbalanced manner. For instance, participant A performed maximum hopping, followed by 2 Hz hopping, and then preferred hopping. However, participant B performed preferred hopping first, followed by maximum hopping, and finally 2 Hz hopping.

Line 114: Change “your” with “their”.

Answer to the reviewer: Changed

Line 119: Authors wrote: “All three hopping were performed pairs of legs.” Please, clarify the sentence.

Answer to the reviewer: Thank you for your attention, more details added, line 139-140

Lines 119-120: I am sure that basketball players with more than 3 years of training experience are familiar with vertical jumping tests. However, Authors should clarify whether a familiarization training session was performed prior to the three jumping conditions or prior to fatigue training protocol.

Answer to the reviewer: As described in line 130, the athletes were allowed to practice at the same assessment session and were explained the procedure for conducting the test.

Fatigue protocol

Authors should clarify the sets, repetitions and rest intervals of the training protocol. Moreover, the fatigue protocol contains exercises with external loads (medicine balls and weighted plates). Define the loads used for each player (load/body mass) and explain how all players underwent the same fatigue prior to the vertical jump testing.

 Answer to the reviewer: Thank you so much for your attention,

It is clarified in the methodology section.

To ensure core muscle fatigue, the subjects were asked to continue exercise to the point of failure. According to previous studies, two criteria were used to define the core muscle fatigue [19,20]: 1-When the subject unable to do exercise in the fourth set properly or correct form. 2. When the subject unable to do exercise with a repetition rate in two seconds in the fourth set.

Methodology: How many times subjects performed the fatigue protocol? I assume that this was performed three times equal to the three different vertical jumping tests.

Answer to the reviewer: The duration of the protocol was about 30 minutes and four sets for each exercise.

Why only the last 5 jumps were used in the analysis?   

Answer to the reviewer: We excluded the initial and final jumps and only calculated the middle 5 jumps to minimize computational errors. In fact, the initial and final jumps might not have been executed completely and correctly by the participant due to the start and end of the movement.

Results:

Table 1: There seemed to missing units of measurement. Also, Authors wrote “* Statistically significant correlation (P ≤ 0.05).”, but no correlation is presented.  Where there any differences between preferred, 2.2Hz or Maximum conditions?

Answer to the reviewer: Only significance in the pre- and post-test has been examined and is displayed with *

Figure 1: Is figure 1 presents the same results as table 1?

Answer to the reviewer: Yes, Figure 1 is a graph of the Ground reaction forces used to calculate the stiffness values presented in Table 1. Also, Figure 1 presents data from one participant, while Table 1 presents group data.  

Discussion:

Discussion has the same problem with introduction. Needs paragraphs (see my comment above).

Answer to the reviewer: Amended, as requested.

Is the fatigue protocol functional and real-world for basketball players?  

Answer to the reviewer: Yes, It has been explain in the introduction

Lines 177-180: Why only stiffness was measured and not the actual vertical cm height of the jumps? This might have provided further kinematical details into the results.

Answer to the reviewer: The aim of our study was not to investigate the lower limb stiffness. Instead, the objective of this study was to examine the lower limb stiffness using force plate and kinetic data.

Line 196: Provide the explanation of ROM before using the abbreviation.

Answer to the reviewer: Changed.

Lines 232-234: This should be in methods.

Answer to the reviewer: The sentence moved to the methodology.

Limitations are missing. Authors should add a paragraph of limitations.

Answer to the reviewer: Done.

Discussion is to general and not discussing the findings of the study. A few lines are focusing on the actual explanations of the results.

Answer to the reviewer: The discussion has been improved and it is highlighted

Conclusions paragraph is confusing, and my suggestion is to be written again focusing on the findings of the study.  

Answer to the reviewer: The conclusion has been rewritten, it can be see on lines 302-308 and can be read

“ According to the results of this study, there was a significant decrease in the amount of lower extremity stiffness measured in various frequencies, following a fatigue protocol targeting the trunk stabilizing muscles, applied in a group of healthy basketball players. Training programs with the aim to increase the endurance enhancement of the stabilizing muscles of the trunk may lead to less severe deteriorations of lower limb stiffness to improve performance as well as to act as an injury prevention mechanism for safe participation in various athletic activities involving the lower limb”.

Reviewer 2 Report

I have analyzed the study "The effects of core stabilization muscles fatigue on lower limb stiffness of Basketball players" and I would like to suggest some suggestions and questions to the authors.

1) Can you explain what you mean in line 47 that the Gluteus Medium "is the core muscle"?

2) In line 58 you attribute to reference 10 the "definition" of stiffness, are you sure?

3) Reference 13 and many others are missing from the DOI

4) Why is the figure of "only one" participant inserted in line 158?

5) I did not find the test-retest reliability performed between subjects

6) I have not found the limitations to the study

I look forward to being able to better evaluate the study after the necessary changes

Author Response

We would like to thank the editor and reviewers for their valuable comments; they have improved the quality of our study We diligently addressed the comments and look forward to clarifying further as needed. Manuscript changes in green color and our responses are the following:

Comments and Suggestions for Authors

1) Can you explain what you mean in line 47 that the Gluteus Medium "is the core muscle"?

Answer to the reviewer: Corrected that with the phrase “more centrally located”.

2) In line 58 you attribute to reference 10 the "definition" of stiffness, are you sure?

Answer to the reviewer: Thank you for your attention, the reference has been changed

Brazier J, Maloney S, Bishop C, Read PJ, Turner AN. Lower Extremity Stiffness: Considerations for Testing, Performance Enhancement, and Injury Risk. J Strength Cond Res. 2019 Apr;33(4):1156-1166. doi: 10.1519/JSC.0000000000002283. PMID: 29112054.

3) Reference 13 and many others are missing from the DOI

Answer to the reviewer: Done.

4) Why is the figure of "only one" participant inserted in line 158?

Answer to the reviewer: Thank you for your attention, the figure caption corrected.

5) I have not found the limitations to the study

Answer to the reviewer: The limitation of the study was added at the end of the discussion, Lin 290-297.

Reviewer 3 Report

General Comments

This is an interesting study, evaluating the effect of core stabilization muscle fatigue on lower limb stiffness during hopping exercise. A total 30 young university level basketball players with a mean age of 21 years participated in this study. The participants performed the hopping test (15 jumps) in three conditions: preferred (no frequency, height or contact time control), maximum, and 2.2 Hz frequency on the force plate. The duration of the core stability muscle fatigue exercise was 30 min. Each workout was performed 20 x 40 sec with rest time between workouts 20 sec and included seated upper torso rotations with medicine ball, static prone torso extension with the medicine ball, supine lower torso rotations with medicine ball, incline sit-ups with weighted plate, lateral side bend (performed bilaterally) with weighted plate, rotating lumbar extension with weighted plate, and standing torso rotations with weighted pulley resistance. The participants were asked to continue exercising to the point of failure. The stiffness of the lower extremities was measured before and after the fatigue protocol. The results of this study clearly indicated reduced lower extremity stiffness in three hopping tests after the core stability muscle fatiguing exercise in young male basketball players.

The manuscript is generally well-written. I would like to suggest some suggestions and questions to the authors

1. I recommend not to duplicate the presentation of results (Table 1 and Figure 2), and, therefore, I suggest to delete Table 1 (there are not presented the unit of measurement of stiffness) and present the main output of this study in Figure 2.  

2. In my opinion, it is obligatory to add limiting factors of this study at the end of the Discussion: (a) only young male university level athletes with unknown qualification were measured and the results of this study cannot transfer to the general population (including women and middle-aged and older subjects) or top athletes with much higher training loads and performance; (b)  body composition (fat %, lean body mass, muscle mass of the lower extremities, etc.) was not assessed and associated with core muscle fatigability and lower limb stiffness during hopping exercise in this study; (c) activation of leg extensor muscles (EMG activity) during stiffness measurement in hopping test before and after the core stability muscle fatigue was not assessed. All these factors cause some difficulties to interpret the results of this study. This notice  should be mentioned and analysed at the end of the Discussion (Limitations).

3. I suggest to rewrite the Conclusions, including the main outcome of this study (based on the results of the study) - core stability muscle fatiguing exercise induced decreased lower extremity stiffness in hopping in young men. In the present form, the Conclusions do not indicate in the best way the effect of core stabilization muscle fatigue on lower limb stiffness during hopping exercise.  

Specific Comments

Abstract

Page 1. Please describe shortly the core stability muscle fatiguing exercise protocol.  

2. Materials and Methods

2.5. Statistical analysis

Page 4. Please indicate that the data are presented as mean ± standard deviation (SD).

3. Results

Page 4. Please delete Table 1 (see General Comments).

Page 5. Please add on Figure 2 capture that data are mean ± SD.

Page 5. Please correct statistical significance between pre- and posttest - * Statistically significant (P ≤ 0.05) compare to pretest, instead of  * Statistically significant correlation (P ≤ 0.05) – this is not correlation.

4. Discussion

Page 7. Please add limiting factors of this study at the end of Discussion (see General Comments).

5. Conclusions

Page 7. Please correct the Conclusions as indicated in General Comments.

References

Page 9. I suggest to delete the next reference, because conference proceedings/abstracts normally not accepted by international refereed journals as valid references:

33. A. Faria, R. Gabriel, J. Abrantes, P. Wood, and H. Moreira, “TRICEPS SURAE MUSCULOTENDINOUS STIFFNESS: RELATIVE DIFFERENCES BETWEEN SOCCER AND NON-SOCCER PLAYERS,” in ISBS-Conference Pr

Author Response

We would like to thank the editor and reviewers for their valuable comments; they have improved the quality of our study We diligently addressed the comments and look forward to clarifying further as needed. Manuscript changes in green color and our responses are the following:

Abstract: it would be interesting to include practical applications

Answer to the reviewer: These are now included, thank you.

Keywords: it is interesting to improve indexing, that the words are different from those of the title

Answer to the reviewer: Some keywords have been changed

INTRODUCTION

There is talk of the specific sport of basketball, but I would like to know if there are studies in other sports such as athletics (other than team sports). Perhaps an introductory paragraph highlighting the importance of abdominal stability in general in sport would be interesting.

Answer to the reviewer: Thank you for your attention. It has been added to the introduction, lines 45-50

I recommend separating the introduction paragraphs as they are all together. In the same way, in line 90 I think there is some space left over

Answer to the reviewer: Amended, thank you.

It is necessary to include the hypotheses of the study

Answer to the reviewer: Included in lines 99-101.

MATERIALS AND METHODS

There is one space left on line 125 and 126

 Answer to the reviewer: Corrected

If possible, it would be interesting to include some images (anonymized) of how the fatigue protocol was done.

Answer to the reviewer:  Thank you so much.  The quality of available pictures is not well

2.5. Statistical analysis

the source of the software must be indicated.

Answer to the reviewer: SPSS v. 22 was used.

In turn, it should be indicated how the data was normalized, why were parametric procedures used?,

Answer to the reviewer: Thank you for your attention, the following sentence added

Answer to the reviewer: The Shapiro-Wilk test results indicated that all data normally distributed

In turn, did all the subjects included in the study comply with the entire inclusion procedure?

Answer to the reviewer: Based on the inclusion criteria, all participants demonstrated compliance with the established procedure.

RESULTS

They are not too elaborate, since you don't see too much information to be able to extract either. I consider the figure could be described a little more

Answer to the reviewer: Thank you so much, more information added to the figure capture

DISCUSSION

Start with the objective and the study hypothesis. As in the introduction, I consider it is important to divide the paragraphs to better understand the explanation.

Answer to the reviewer: Corrected.

Limitations such as the sample (it is not very large, they are only men, were they from similar or different clubs?, etc.) and possible future researchers such as interventions to improve abdominal stability or consider sports performance parameters should be included.

Answer to the reviewer: Corrected.

REFERENCES

They should be thoroughly reviewed following the journal's rules.

Answer to the reviewer:  Thank you so much. The reference style is updated according to the journal format

Reviewer 4 Report

Dear authors

Thanks for this interesting article. I would like to suggest some suggestions and questions to the authors.

Abstract: it would be interesting to include practical applications

Keywords: it is interesting to improve indexing, that the words are different from those of the title

INTRODUCTION

There is talk of the specific sport of basketball, but I would like to know if there are studies in other sports such as athletics (other than team sports). Perhaps an introductory paragraph highlighting the importance of abdominal stability in general in sport would be interesting.

I recommend separating the introduction paragraphs as they are all together. In the same way, in line 90 I think there is some space left over

It is necessary to include the hypotheses of the study

MATERIALS AND METHODS

There is one space left on line 125 and 126

If possible, it would be interesting to include some images (anonymized) of how the fatigue protocol was done.

2.5. Statitiscal analysis: the source of the software must be indicated. In turn, it should be indicated how the data was normalized, why were parametric procedures used?, in turn, did all the subjects included in the study comply with the entire inclusion procedure?

RESULTS

They are not too elaborate, since you don't see too much information to be able to extract either. I consider the figure could be described a little more

DISCUSSION

Start with the objective and the study hypothesis. As in the introduction, I consider it is important to divide the paragraphs to better understand the explanation.

Limitations such as the sample (it is not very large, they are only men, were they from similar or different clubs?, etc.) and possible future researchers such as interventions to improve abdominal stability or consider sports performance parameters should be included.

REFERENCES

They should be thoroughly reviewed following the journal's rules.

Author Response

We would like to thank the editor and reviewers for their valuable comments; they have improved the quality of our study We diligently addressed the comments and look forward to clarifying further as needed. Manuscript changes in green color and our responses are the following:

  1. I recommend not to duplicate the presentation of results (Table 1 and Figure 2), and, therefore, I suggest to delete Table 1 (there are not presented the unit of measurement of stiffness) and present the main output of this study in Figure 2.

Answer to the reviewer:  Considering that the table effectively presents the means and standard deviations of the results and enhances readability, we opt to retain the table.

  1. In my opinion, it is obligatory to add limiting factors of this study at the end of the Discussion: (a) only young male university level athletes with unknown qualification were measured and the results of this study cannot transfer to the general population (including women and middle-aged and older subjects) or top athletes with much higher training loads and performance; (b) body composition (fat %, lean body mass, muscle mass of the lower extremities, etc.) was not assessed and associated with core muscle fatigability and lower limb stiffness during hopping exercise in this study; (c) activation of leg extensor muscles (EMG activity) during stiffness measurement in hopping test before and after the core stability muscle fatigue was not assessed. All these factors cause some difficulties to interpret the results of this study. This notice should be mentioned and analyzed at the end of the Discussion (Limitations).

Answer to the reviewer: Amendments made accordingly.

  1. I suggest to rewrite the Conclusions, including the main outcome of this study (based on the results of the study) - core stability muscle fatiguing exercise-induced decreased lower extremity stiffness in hopping in young men. In the present form, the Conclusions do not indicate in the best way the effect of core stabilization muscle fatigue on lower limb stiffness during hopping exercise.

Answer to the reviewer: Amendments made accordingly.

 Specific Comments

Abstract

Page 1. Please describe shortly the core stability muscle fatiguing exercise protocol.

 Answer to the reviewer: This part has been improved

  1. Materials and Methods

2.5. Statistical analysis

Page 4. Please indicate that the data are presented as mean ± standard deviation (SD).

Answer to the reviewer: Added

  1. Results

Page 4. Please delete Table 1 (see General Comments).

Answer to the reviewer: Considering that the table effectively presents the means and standard deviations of the results and enhances readability, we opt to retain the table.

Page 5. Please add on Figure 2 capture that data are mean ± SD.

Answer to the reviewer: It has been added

Page 5. Please correct statistical significance between pre- and posttest - * Statistically significant (P ≤ 0.05) compare to pretest, instead of * Statistically significant correlation (P ≤ 0.05) – this is not correlation.

Answer to the reviewer: Amended.

  1. Discussion

Page 7. Please add limiting factors of this study at the end of Discussion (see General Comments).

Answer to the reviewer: Added.

  1. Conclusions

Page 7. Please correct the Conclusions as indicated in General Comments.

Answer to the reviewer: Amended, thank you.

References

Page 9. I suggest to delete the next reference, because conference proceedings/abstracts normally not accepted by international refereed journals as valid references:

  1. A. Faria, R. Gabriel, J. Abrantes, P. Wood, and H. Moreira, “TRICEPS SURAE MUSCULOTENDINOUS STIFFNESS: RELATIVE DIFFERENCES BETWEEN SOCCER AND NON-SOCCER PLAYERS,” in ISBS-Conference Pr

Answer to the reviewer: Thank you for attention, the reference has been changed

Round 2

Reviewer 1 Report

No comments

Reviewer 2 Report

Dear authors,

I have carefully read your responses to my queries. I believe that the revisions you have made and the improvements are perfectly in line with my expectations. Therefore, I see no limitations to publication in this latest version provided.

Best regards

Reviewer 4 Report

Thank you for your collaboration to improve this paper